# Benefit of Neoadjuvant Laparoscopic Hyperthermic Intraperitoneal Chemotherapy and Bidirectional Chemotherapy for Patients with Gastric Cancer with Peritoneal Carcinomatosis Considering Cytoreductive Surgery

**DOI:** 10.3390/cancers15133401

**Published:** 2023-06-29

**Authors:** Hsin-Hsien Yu, Yutaka Yonemura, Hui-Ji Ng, Ming-Che Lee, Bor-Chyuan Su, Mao-Chih Hsieh

**Affiliations:** 1Division of General Surgery, Department of Surgery, School of Medicine, College of Medicine, Taipei Medical University, Taipei 116, Taiwan; klaus610@gmail.com (H.-H.Y.);; 2Division of General Surgery, Department of Surgery, Wan Fang Hospital, Taipei Medical University, Taipei 116, Taiwan; 3Comprehensive Care Center for Peritoneal Metastasis, Wan Fang Hospital, Taipei Medical University, Taipei 116, Taiwan; 4Peritoneal Dissemination Center, Kishiwada Tokushukai Hospital, Kishiwada 596-8522, Osaka, Japan; 5Department of Surgery, Kusatsu General Hospital, Kusatsu 525-8585, Shiga, Japan; 6Department of Anatomy and Cell Biology, School of Medicine, College of Medicine, Taipei Medical University, Taipei 11031, Taiwan

**Keywords:** hyperthermic intraperitoneal chemotherapy, bidirectional chemotherapy, gastric cancer, peritoneal carcinomatosis, cytoreductive surgery

## Abstract

**Simple Summary:**

The comprehensive treatment of using neoadjuvant laparoscopic HIPEC and bidirectional chemotherapy before cytoreductive surgery has been introduced. We reported the real benefit of this strategy and analyze the prognostic factors on outcome. We also provided a recommended patient selection criteria for applying this protocol.

**Abstract:**

Comprehensive treatment comprising neoadjuvant laparoscopic HIPEC (L-HIPEC) and bidirectional intraperitoneal and systemic induction chemotherapy (BISIC) followed by cytoreductive surgery (CRS) for gastric cancer with peritoneal carcinomatosis (PC) has been developed. However, its benefits and patient selection criteria have not been thoroughly investigated. We retrospectively reviewed 113 patients, with 25 having received comprehensive treatment (L-HIPEC, BISIC, and then CRS-HIPEC; the BISIC group) and 88 having received direct CRS-HIPEC (the CRS group). The BISIC group showed greater tumor clearance in terms of post-CRS peritoneal cancer index ((PCI) 6 vs. 14, *p* = 0.002) compared to CRS group. The median survival was 20.0 months in the BISIC group and 8.6 months in the CRS group (*p* = 0.031). Multivariable analysis revealed that the factors associated with increased survival were the BISIC protocol, age, and post-CRS tumor clearance. BISIC significantly improved survival in cases of moderate severity (PCI 11–20) and severe cases (PCI 21–39) without increasing the morbidity rate. We recommend the use of this neoadjuvant strategy for patients with gastric cancer-associated PC and an initial PCI of >10 to provide superior survival outcomes.

## 1. Introduction

Gastric cancer is the fifth most common cancer and the fourth leading cause of cancer deaths worldwide [1]. Approximately 17% of patients with gastric cancer have peritoneal carcinomatosis (PC) at the time of diagnosis, and this proportion is 40% among patients with advanced gastric cancer [2]. Even after curative surgery, PC recurrence affects nearly 43% of patients with gastric cancer regardless of cancer stage [3]. PC is considered a terminal illness, and its prognosis remains poor despite new chemotherapeutic drugs and agents with new molecular targets. Currently, systemic therapy or best supportive care is the standard recommended treatment for this condition. Since 2013, studies have reported a median survival of only 8–13.2 months [4]. In the late 1990s, the Peritoneal Surface Oncology Group International (PSOGI) introduced a novel treatment, cytoreductive surgery (CRS) combined with intraoperative hyperthermic intraperitoneal (IP) chemotherapy (HIPEC), for peritoneal carcinomatosis originating from various cancer types [5]. The aim of cytoreductive surgery is to eradicate macroscopic tumors, and the purpose of HIPEC is to eliminate tiny or microscopic seeding tumors. CRS-HIPEC has exhibited survival benefits in the treatment of PC from malignant mesothelioma and gynecological and gastrointestinal cancers [6,7,8]. Yonemura et al. reported a median survival of 15.5 months in patients with gastric cancer with peritoneal carcinomatosis in which complete cytoreduction (CC) was achieved and 7.9 months in those in which cytoreduction was incomplete [9]. In a French multi-institutional study, Glehen et al. reported 5-year survival rates of 23% for such patients after CC and 13% for those after incomplete cytoreduction [10].

Although the clinical results of CRS-HIPEC for PC from gastric cancer have been promising, the prognosis of patients with gastric cancer with peritoneal carcinomatosis with a large tumor burden remains poor, and achievement of complete cytoreduction in these patients is difficult. In the mid-2010s, PSOGI developed a comprehensive gastric cancer with peritoneal carcinomatosis treatment strategy including CRS with chemotherapy for reducing large tumor burdens before surgery [11]. The procedures include a first laparoscopic exam for a tissue biopsy, peritoneal cancer index (PCI) evaluation, and intraoperative HIPEC followed by neoadjuvant bidirectional chemotherapy and interval CRS-HIPEC with early postoperative IP chemotherapy and late systemic chemotherapy [12]. Yonemura et al. reported a 5-year survival rate of 32.4% for patients who responded to neoadjuvant chemotherapy without malignant ascites and with a PCI lower than a cutoff of 7, and who had received complete CRS [12].

However, few studies have reported the long-term results of this structured comprehensive strategy. The indications for this comprehensive treatment and the benefits of neoadjuvant laparoscopic HIPEC (L-HIPEC) and bidirectional chemotherapy before CRS have also not been clarified. Thus, this study assessed real-world survival data to investigate the effectiveness of this strategy. In addition, this study explored the indications for this treatment and whether an adequate cutoff level exists indicating that direct CRS should be avoided.

## 2. Materials and Methods

### 2.1. Patients

The data for the current retrospective observational study were collected from a prospectively maintained database of patients receiving CRS and HIPEC for gastric cancer with peritoneal carcinomatosis at a comprehensive care center for patients with peritoneal metastasis of an Asian hospital between January 2005 and December 2020. The inclusion criteria were pathologically confirmed gastric cancer with PC or malignant ascites, Eastern Cooperative Oncology Group performance status of 0–2, medical fitness for major cytoreductive surgery at any stage of treatment, and the absence of extra-abdominal metastasis. The exclusion criteria were HIPEC with prophylactic intent for locally advanced gastric cancer without PC, HIPEC with palliative intent for only ascites control or symptom relief, uncontrolled comorbidities, and a lack of medical fitness for major operations. In total, 113 patients were enrolled. Among them, 25 received initial laparoscopic examination with L-HIPEC and 3 courses of bidirectional IP and systemic induction chemotherapy (BISIC) following protocol described previously [12,13,14]. Interval CRS and secondary HIPEC were performed on these patients (the BISIC group). Another 88 patients received CRS and direct HIPEC without neoadjuvant treatment (the CRS group). The choice of BISIC or CRS group was determined by the shared decision-making mode between multidisciplinary team and patients as well as with patients’ families after the evaluation of disease severity and an explanation of the current evidence of benefit in each treatment protocol by physicians. The institutional review board of the study hospital approved this study (TMU-JIRB-C N201807067).

### 2.2. Study Design

The study protocol and grouping are illustrated in Figure 1. Patients in the BISIC group received laparoscopic examination for tissue biopsy and evaluation of tumor burden using the PCI as described by Sugarbaker [15,16]. The PCI is based on scores for 13 abdominal regions, where scores of 0, 1, 2, and 3 indicate no tumor and tumors sized <0.5, 0.5–5, and >5 cm, respectively; the PCI is the sum of the score for of all 13 regions and thus ranges from 0 to 39. L-HIPEC was performed for 60 min using the closed technique with 6 L of saline, and IP chemotherapeutic drugs were administered at 42 °C throughout the procedure. Cisplatin (30 mg/m^2^) and docetaxel (30 mg/m^2^) were used as the chemotherapeutic drugs for HIPEC. After the procedure, a Tenckhoff catheter (peritoneal dialysis tubing system) was implanted for later bidirectional chemotherapy. After 3 weeks of recovery from the preceding surgery, the patients received IP and systemic intravenous (IV) chemotherapy with cisplatin and docetaxel administered on days 1 and 8, respectively, during each 3-week course. Systemic chemotherapy with oral TS-1 (gimeracil, oteracil, tegafur; Taiho Pharmaceutical, Tokyo, Japan) was administered for 14 days every 3 weeks at a dosage of 120 mg per day for a body surface area (BSA) of >1.5 m^2^, 100 mg for a BSA of 1.25–1.5 m^2^, or 80 mg for a BSA of <1.25 m^2^, in divided doses. After 3 courses of BISIC, the patients received CRS and secondary HIPEC.

The CRS was performed in an attempt to remove all visible tumors and organ tissues affected by malignancy through standard peritonectomy, as described by Sugarbaker [15]. After the CRS, the extent of tumor clearance was determined on the basis of PCI and cytoreduction scores calculated according to criteria described by Sugarbaker: cytoreduction scores of 0, 1, 2, and 3 indicated no residual tumor and residual tumors sized <2.5, 2.5–25, and >25 mm, respectively [16]. Subsequently, the secondary HIPEC procedure was performed, and all patients were closely monitored in intensive care units postoperatively. Morbidity and mortality were graded on the basis of the Clavien–Dindo classification system [17]. The patients received standard adjuvant systemic surgery and routine follow-up in our outpatient clinic, including monitoring of tumor markers and computed tomography, magnetic resonance imaging, or positron emission tomography-computed tomography to detect disease recurrence.

### 2.3. Statistical Analysis

Survival duration (to death or final follow-up) was evaluated through Kaplan–Meier analysis and compared among subgroups using a log-rank test to identify significant differences. Univariable and multivariable Cox proportional hazards regressions were used to assess the associations between potential predictors and overall survival after CRS-HIPEC. Categorical variables were compared using the chi-square or Fisher’s exact test. Continuous variables were analyzed using Student’s *t*-test. Statistical analyses were performed using SPSS version 23.0 (IBM Analytics, Armonk, NY, USA). A *p* value of <0.05 indicated significance.

## 3. Results

### 3.1. Patient Demographics

The demographics of the study population are presented in Table 1. Between January 2005 and December 2020, 113 patients with diagnoses of gastric cancer with peritoneal carcinomatosis underwent treatment with curative intent at our institute. The average age was 53.1 years, and age ranged from 29 to 79 years. Of the patients, 42 were men and 71 were women. Among these patients, 25 received neoadjuvant BISIC before CRS-HIPEC and 88 received CRS-HIPEC directly. Patients in these two groups did not differ significantly in age, body mass index, the preoperative use of total parenteral nutrition, American Society of Anesthesiologists physical status score, or heart function (ejection fraction of cardiac contraction). The average levels of tumor markers were high: carcinoembryonic antigen at 12.1 ng/mL, cancer antigen 19-9 at 555.8 U/mL, and cancer antigen 125 at 83.8 U/mL, but these did not differ significantly between the groups. Only a lower platelet count in the BISIC group (196.0 vs. 279.4 per 103/μL, *p* = 0.001) was noted before CRS-HIPEC than in the CRS group because of the effect of the neoadjuvant chemotherapy.

### 3.2. Perioperative Outcomes of Cytoreductive Surgery

The PCIs before CRS in the BISIC and CRS groups were 16 and 21 (*p* = 0.017), respectively, with the main score differences relating to the small-bowel mesentery (PCI regions 9–12; 3 vs. 6, *p* = 0.009; Table 2). Completeness of cytoreduction (CC) score of 0 or 1 was achieved in 60% of patients in the BISIC group and 44.3% of patients in the CRS group (*p* = 0.329). Although the cytoreduction score did not differ significantly, another more detailed indicator, postoperative PCI, differed significantly: it was 6 in the BISIC group and 14 in the CRS group (*p* = 0.002). This indicates that the BISIC resulted in greater tumor clearance after CRS. Although operation time, blood loss, and the number of transfusion units were higher in the BISIC group, the postoperative morbidity rate did not differ from that in the CRS group. The complication events of cytoreductive surgery in the BISIC group included anastomosis leakage or bowel perforation (*n* = 2), wound infection (*n* = 1), pneumonia (*n* = 3), and acute kidney injury (*n* = 1), whereas in the CRS group, these included anastomosis leakage or bowel perforation (*n* = 15), wound infection or poor healing (*n* = 12), pneumonia (*n* = 7), acute kidney injury (*n* = 2), hemorrhage (*n* = 2), pancreatic leakage (*n* = 1), hepatic failure (*n* = 1), and splenic infarction (*n* = 1). The cause of surgical mortality in the BISIC group was anastomosis leakage related sepsis (*n* = 1), whereas in the CRS group, the causes of surgical mortality were anastomosis leakage-related sepsis (*n* = 4), pneumonia with respiratory failure (*n* = 3), and hepatic failure (*n* = 1).

### 3.3. Survival Outcomes and Multivariable Analysis of Predictive Factors

The median survival time (MST) after enrollment was 20.0 months in the BISIC group and 8.6 months in the CRS group (*p* = 0.031; Figure 2a). The “survival time after enrollment” refers to the survival period calculated from the time of initial visit. The 1-, 2-, and 5-year overall survival rates for the BISIC and CRS groups were 83% and 43%, 35% and 25%, and 9% and 9%, respectively, and differed significantly (*p* = 0.031; Table 3). Thus, BISIC significantly enhanced overall survival. MST after CRS did not differ significantly between the groups, indicating that the improvement in overall survival resulted from BISIC treatment (Figure 2b). To further determine factors related to prognosis in our study patients, we performed Cox proportional hazards regression. The results revealed that the factors significantly related to superior survival were the BISIC protocol (*p* = 0.049), patient age (*p* = 0.013), and post-CRS residual tumor burden (i.e., PCI; *p* = 0.002; Table 4).

### 3.4. Selection Criteria for BISIC Based on PC Grade

The ideal criteria for selecting between neoadjuvant L-HIPEC with the BISIC protocol before CRS and direct CRS are unclear. To examine survival differences related to tumor burden, we stratified the patients by initial PCI into mild (1–10), moderate (11–20), and severe (21–39) subgroups. The results indicated inferior survival for direct CRS for patients with moderate or severe PC (i.e., PCI > 10; Table 5). BISIC improved survival for patients with moderate PC (30.0 vs. 23.3 months, *p* = 0.029) and PC of any severity (BISIC vs. CRS group, 19.0 vs. 6.6 months; *p* = 0.015; Figure 3). These results suggest that neoadjuvant BISIC is advisable for patients with an initial PCI of >10.

### 3.5. Oncological Effects of BISIC

Neoadjuvant chemotherapy is used for reducing tumor burden before major tumor resection. Such treatment is effective in cancers of various systems. To determine the efficacy of neoadjuvant BISIC for gastric cancer with peritoneal carcinomatosis, we compared the tumor burden and ascites cytology of patients before and after neoadjuvant BISIC prior to CRS (Table 6). BISIC reduced the mean PCI from 27 ± 2.3 to 16 ± 2.1 (*p* = 0.003) and median PCI from 27 to 14 (Figure 4). In addition, malignant ascites was noted in 12 patients before BISIC but in only 8 patients afterward. The rate of conversion of malignant ascites was 33%. No malignant ascites was noted after BISIC in patients free from malignant ascites initially. This result indicates the oncological efficacy of BISIC.

## 4. Discussion

Systemic chemotherapy is the standard treatment for stage IV gastric cancer. However, the prognosis of patients with peritoneal metastasis is worse than that of those with other distant metastases [18]. CRS-HIPEC improves the outcomes of PC with several specific etiologies, including appendix and ovarian cancer and mesothelioma. A randomized phase III study by Yang et al. [19]. demonstrated that CRS-HIPEC was associated with longer disease-free survival than CRS alone (11.0 vs. 6.5 months, *p* < 0.05) in patients with gastric cancer with peritoneal carcinomatosis. A randomized phase III study, the GYMSSA trial, demonstrated that CRS-HIPEC was associated with longer overall survival than systemic therapy alone (11.3 vs. 4.3 months), but statistical significance was not reached because of limited patient numbers [20]. A case–control propensity-score matching study, the Cytoreductive Surgery With or Without Hyperthermic Intraperitoneal Chemotherapy for Gastric Cancer With Peritoneal Metastases study, investigated 277 patients from 19 French centers and demonstrated that CRS-HIPEC was associated with longer overall survival that CRS alone (18.8 vs. 12.1 months, *p* < 0.01) [8]. The study demonstrated not only the effectiveness of HIPEC after CRS but also the crucial role in survival improvement of CC to a score of 0 or 1. Recently, the Italian Peritoneal Surface Malignancies Oncoteam (S.I.C.O), with 11 high-volume, specialized centers, has also demonstrated that the survival time of gastric cancer after CRS-HIPES was 20.2 months. In this Italian multi-center study, they revealed that the significantly better prognostic factors were PCI score ≤ 6 (44.3 vs. 13.4 months, *p* = 0.005), CC score 0 (40.7 vs. 10.7 months, *p* = 0.003), and receiving neoadjuvant chemotherapy (35.3 vs. 10.7 months, *p* = 0.022) [21]. In guidelines for the treatment of many advanced cancers, including ovarian, breast, pancreatic, rectal, and gastric cancer, neoadjuvant chemotherapy with or without radiotherapy is a standard strategy. Expectational downstaging or the conversion of inoperable conditions to operable conditions improves tumor clearance and long-term outcomes. Because of persistently poor results and only limited prognosis improvement in patients with gastric cancer with peritoneal carcinomatosis treated with CRS-HIPEC, in 2006, Yonemura introduced neoadjuvant IP and systemic chemotherapy (NIPS) using IP cisplatin and docetaxel on days 1 and 8 with systemic oral S-1 on days 1–14 every 3 weeks [22]. The concept of IP chemotherapy with directly contacting peritoneal seeding tumors is based on consideration of the blood–peritoneum barrier, which is the stromal tissue between the mesothelium and submesothelial capillaries and impedes the transport of drugs from the systemic circulation to the IP cavity [23]. In 2014, Yonemura modified this treatment to include BISIC using IP and IV cisplatin plus docetaxel on days 1 and 8 with systemic oral S-1 on days 1–14 every 3 weeks for 3 courses [13]. Compared with NIPS, BISIC was associated with a higher histological response rate (60% vs. 83%) and conversion rate of malignant to nonmalignant ascites (70% vs. 79%) [12]. In 2016, PSOGI developed a novel comprehensive treatment for gastric cancer with peritoneal carcinomatosis with curative intent involving laparoscopic examination with neoadjuvant HIPEC, neoadjuvant BISIC, CRS-HIPEC, and postoperative adjuvant chemotherapy [11].

Real-world data on the use of this strategy are limited, and the benefits of neoadjuvant L-HIPEC and BISIC before CRS are unclear. In the present study, although the initial mean PCI was greater in the BISIC group than in the CRS group (27 vs. 21, *p* = 0.045), the mean PCI was lower after BISIC treatment 16 vs. 21, *p* = 0.017; Table 2). In the BISIC group, the mean PCI significantly improved from 27 to 21 (*p* = 0.003), and the rate of conversion from positive to negative cytology was 33% after the neoadjuvant treatment (Table 6). The clearance rate measured in terms of postoperative PCI was also higher in the BISIC group than in the CRS group (6 vs. 14, *p* = 0.002). The post-CRS PCI was also lower for the most difficult area for tumor clearance (the small-bowel mesentery) in the BISIC group than in the CRS group (3 vs. 5, *p* = 0.025). This demonstrates the effectiveness of neoadjuvant L-HIPEC and BISIC treatment. The difference in overall survival indicated by the Cox multivariate proportional hazards regression suggested the benefit of BISIC (hazard ratio, 0.541; 95% confidence interval, 0.293–0.998, *p* = 0.049). The safety of this treatment also warrants consideration. In a previous study, IP docetaxel achieved a high concentration over 12 h at a low plasma concentration, which resulted in less severe side effects than conventional systemic chemotherapies [24,25]. Yonemura et al. also demonstrated an acceptable side effect rate of 2.6% for grade 3 or 4 hematological events and 3.8% for renal toxicity [26]. In our study, the groups’ pre-CRS conditions were identical except for the lower platelet count in the BISIC group (196.0 vs. 279.4/103/µL, *p* = 0.001; Table 1). Longer operation time, greater blood loss, and the transfusion of more units were observed in the BISIC group. However, these differences did not influence the morbidity rate after CRS (Clavien–Dindo class: 0–2, 68% vs. 67%; class 3 or 4, 28% vs. 23.9%, *p* = 0.680; Table 2). In addition, surgeons may be concerned about the formation of intra-abdominal adhesions, which may prevent future CRS, but many antiadhesive powders, fluids, and films can be used to reduce adhesions.

In our multivariable survival analysis, the key factors influencing survival were age, the use of BISIC, and the clearance rate of small-bowel mesentery seeding tumors (as indicated post-CRS PCI in regions 9 to 12; hazard ratio: 1.218, 95% confidence interval: 1.078–1.376, *p* = 0.002; Table 4). MST differed significantly between the BISIC group (20.0 months) and CRS group (8.6 months; *p* = 0.031; Table 3). However, MST after CRS did not differ significantly (BISIC vs. CRS: 10.9 vs. 8.6 months, *p* = 0.747). This may indicate that L-HIPEC and BISIC can prolong survival. Furthermore, the 5-year survival rates of the groups were similar, at approximately 9%. Few studies have examined the factors affecting the long-term survival of patients with gastric cancer with peritoneal carcinomatosis treated with CRS-HIPEC. Brandl et al. and PSOGI conducted a multi-institutional cohort study using questionnaires and revealed that the factors affecting long-term survival were CC to a score of 0 and a PCI of <6 [27]. Yonemura also demonstrated that the predictors of long-term survival were a PCI of <14 and ascites with nonmalignant ascites [28].

The optimal criteria for selecting the L-HIPEC and BISIC protocol remain unclear. In the present study, the study population was stratified by initial PCI into mild (1–10), moderate (11–20), and severe (21–39) PC groups. The MSTs of the BISIC and CRS groups were 20.0 and 10.8 months (*p* = 0.061) for mild PC, 30.0 and 13.3 months (*p* = 0.029) for moderate PC, and 19.6 and 6.6 months (*p* = 0.015) for severe PC. This result suggests that direct CRS-HIPEC should be avoided for patients with gastric cancer with peritoneal carcinomatosis and a PCI of >10. To increase survival in gastric cancer with peritoneal carcinomatosis, we suggest L-HIPEC followed by 3 BISIC courses before CRS in patients with an initial PCI of >10.

The limitations of this study include the low patient number, use of a single specialized center, and nonrandomized retrospective cohort analysis. However, we standardized the treatment protocol, including the BISIC and HIPEC regimens, temperature, and the duration of HIPEC, and controlled for the quality of surgeon performance in CRS, which is not possible in multicenter studies. Furthermore, propensity-score matching was not performed between the BISIC and CRS groups, and the BISIC group had a higher mean PCI prior to treatment. However, although the patients in the BISIC group exhibited more severe PC, they nonetheless exhibited longer survival after enrollment. Finally, the indications for CRS, notably the cutoff PCI after BISIC, remain controversial. Further multi-institutional randomized-controlled studies with more patients are necessary to clarify these.

## 5. Conclusions

The management of patients with gastric cancer with peritoneal carcinomatosis remains challenging. This study revealed that the application of the comprehensive PSOGI treatment protocol with curative intent effectively improved patient outcomes. L-HIPEC followed by BISIC reduced the severity of carcinomatosis and increased the conversion rate of malignant ascites and tumor clearance after CRS without increasing the morbidity rate. We recommend use of this neoadjuvant strategy to prolong survival, especially for patients with gastric cancer with peritoneal carcinomatosis and an initial PCI of >10.

## Figures and Tables

**Figure 1 cancers-15-03401-f001:**
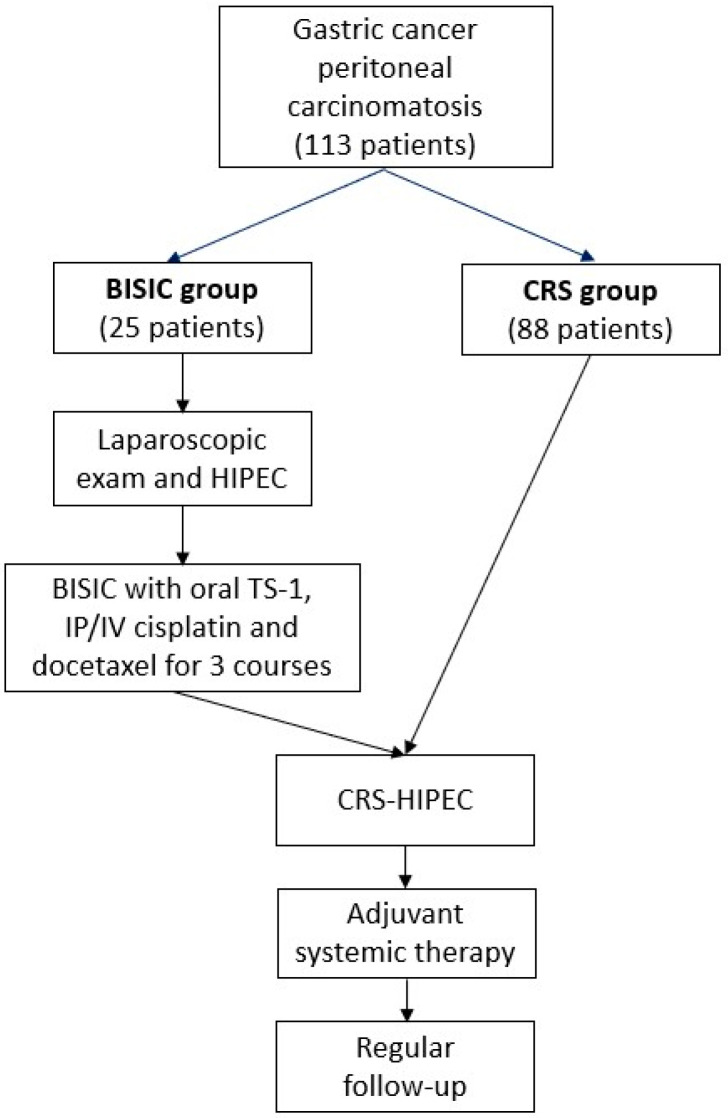
Study design of patients’ grouping and treatments.

**Figure 2 cancers-15-03401-f002:**
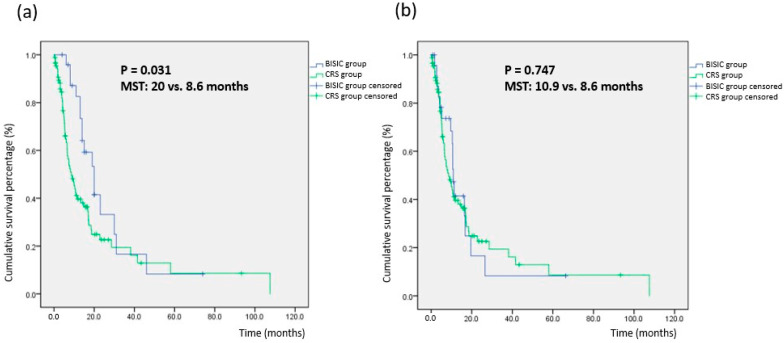
Survival curve comparison between BISIC and CRS group. (**a**) Survival time after enrollment. (**b**) Survival time after final cytoreductive surgery.

**Figure 3 cancers-15-03401-f003:**
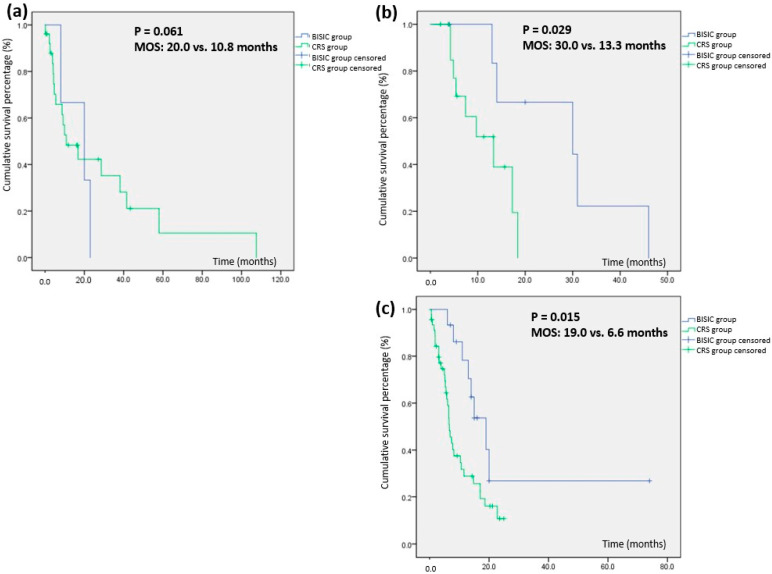
Survival comparison based on grade of carcinomatosis. (**a**) Survival time for patients with initial PCI 1–10, (**b**) survival time for patients with initial PCI 11–20, (**c**) survival time for patients with initial PCI 21–39.

**Figure 4 cancers-15-03401-f004:**
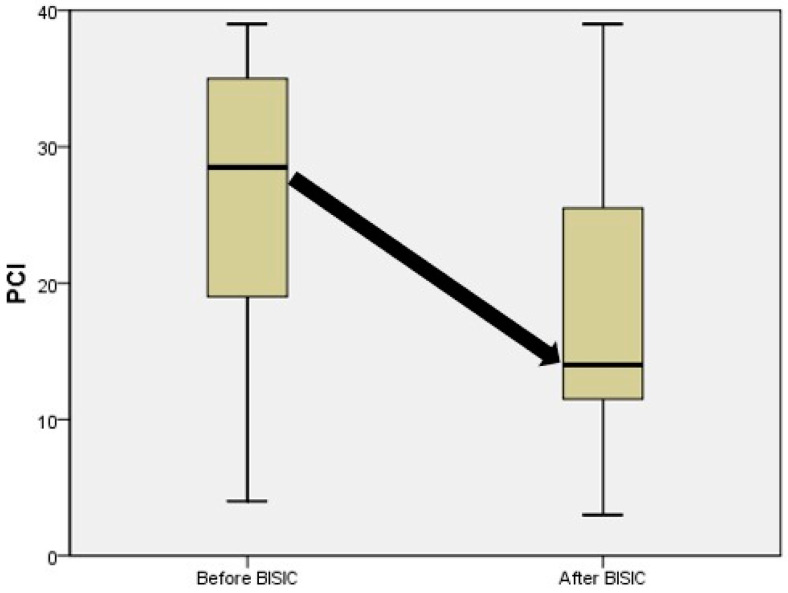
Oncological effects of BISIC treatment. The box and whisker plot showed the alteration of median PCI (black arrow) and shift of middle quartile (25–75%) PCI before and after BISIC.

**Table 1 cancers-15-03401-t001:** Demographics of 113 patients with gastric cancer and peritoneal carcinomatosis.

Variables	All Patients	BISIC Group	CRS Group	*p*-Value
	(*n* = 113)	(*n* = 25)	(*n* = 88)	
Age (years ± SE)	53.1 ± 1.1	50.8 ± 2.3	53.8 ± 1.2	0.232
(range)	(29–79)			
Gender				0.06
Male	42 (37%)	5 (20%)	37 (42%)	
Female	71 (63%)	20 (80%)	51 (58%)	
BMI	21.0 ± 0.4	21.0 ± 0.7	21.1 ± 0.4	0.902
Preoperative TPN	70 (62%)	13 (52%)	57 (65%)	0.254
ASA score				0.778
2	91 (81%)	21 (84%)	70 (80%)	
3	22 (19%)	4 (16%)	18 (20%)	
Ejection fraction of heart (%)	61.28 ± 2.4	56.8 ± 5.7	63.3 ± 2.4	0.296
Laboratory data				
CEA (ng/mL)	12.1 ± 4.1	8.3 ± 3.6	13.3 ± 5.2	0.607
CA 19-9 (U/mL)	555.8 ± 213.3	1097.0 ± 706.7	375.4 ± 159.4	0.329
CA-125 (U/mL)	83.8 ± 14.2	74.3 ± 21.5	87.3 ± 17.9	0.686
AFP	7.2 ± 1.8	5.8 ±1.2	7.8 ± 2.5	0.611
CRP (mg/dL)	1.6 ± 0.4	1.6 ± 0.6	1.6 ± 3.6	0.953
Hb	11.0 ± 0.2	10.4 ± 0.3	11.2 ± 0.2	0.08
PLT (10^3^/μL)	259.6 ± 11.1	196.0 ± 19.0	279.4 ± 12.6	0.001
ALB	3.6 ± 0.1	3.5 ± 0.1	3.6 ± 0.1	0.158

The data was presented with mean ± standard error or percentage (%). *p* < 0.05 indicated a statistical significance. BMI, body mass index; ASA score, American Society of Anesthesiology score; CEA, carcinoembryonic antigen; CA 19-9, carbohydrate antigen 19-9; CA-125, carbohydrate antigen 125; AFP, alpha-fetoprotein; CRP, C-reactive protein; Hb, hemoglobin; PLT, platelet; ALB, albumin.

**Table 2 cancers-15-03401-t002:** Perioperative variables comparison between two study groups.

Variables	All Patients	BISIC Group	CRS Group	*p*-Value
	(*n* = 113)	(*n* = 25)	(*n* = 88)	
Pre-CRS PCI				
Total	20	16	21	0.017
Region 0–8	15	12	15	0.057
Region 9–12	5	3	6	0.009
Post-CRS PCI				
Total	13	6	14	0.002
Region 0–8	8	3	9	0.001
Region 9–12	4	3	5	0.025
CC score				0.329
0	34 (30.4%)	9 (36.0%)	25 (28.4%)	
1	20 (17.9%)	6 (24.0%)	14 (15.9%)	
2	18 (16.1%)	5 (20%)	14 (15.9%)	
3	40 (35.6%)	5 (20%)	35 (39.8%)	
Morbidity	37 (32.7%)	7 (28%)	30 (34%)	0.636
Clavien–Dindo class				0.68
0–2	76 (67.3%)	17 (68.0%)	59 (67.0%)	
3–4	28 (24.7%)	7 (28.0%)	21 (23.9%)	
5	9 (8.0%)	1 (4.0%)	8 (9.1%)	
Peritonectomy	52 (46.0%)	13 (52.0%)	39 (44.3%)	0.506
Operation time (minutes)	506	633 ± 47	466 ± 23	0.001
Transfusion				
PRBC (unit)	4	8	3	0.001
FFP (unit)	3	6	2	
Blood loss (mL)	558 (0–7000)	1131	377	0.019
Length of stay (days)	20 ± 22	19 ± 3	20 ± 4	0.848

The data was presented with number with percentage (%) or mean ± standard error. *p* < 0.05 indicated a statistical significance. PCI, peritoneal cancer index; CC score, completeness of cytoreduction score; PRBC, packed red blood cells; FFP, fresh frozen plasma.

**Table 3 cancers-15-03401-t003:** Survival outcome during follow-up.

Variables	All Patients	BISIC Group	CRS Group	*p*-Value
	(*n* = 113)	(*n* = 25)	(*n* = 88)	
Median survival time after enrollment (months)		20	8.6	0.031
Median survival time after CRS (months)		10.9	8.6	0.747
Overall survival rate (%) after first HIPEC				0.031
1-year	52%	83%	43%	
2-year	27%	35%	25%	
3-year	20%	17%	22%	
4-year	12%	9%	14%	
5-year	9%	9%	9%	

The data was presented by survival percentage and analyzed by log-rank test for comparison. *p* < 0.05 indicated a statistical significance.

**Table 4 cancers-15-03401-t004:** Cox proportional hazard model of survival analysis.

Risk Factor	Hazard Ratio (95% CI)	*p*-Value
Group (BISIC vs. CRS)	0.541 (0.293–0.998)	0.049
Gender (male vs. female)	1.028 (0.607–1.739)	0.919
Age	1.027 (1.006–1.049)	0.013
Initial PCI 9–12	0.939 (0.827–1.065)	0.939
Post-CRS remnant PCI 9–12	1.218 (1.078–1.376)	0.002

Post-CRS remnant PCI 9–12 indicates the remnant PCI of region 9–12 after final cytoreductive surgery.

**Table 5 cancers-15-03401-t005:** Survival differences according to grades of PC.

First Encounter PCI	Median Survival Time	*p*-Value	Range (Overall)
1–10 (*n* = 29)		0.061	
BISIC group (*n* = 3)	20		0.8–39.2
CRS group (*n* = 26)	10.8		0.3–21.3
11–19 (*n* = 23)		0.029	
BISIC group (*n* = 7)	30		0.1–61.4
CRS group (*n* = 16)	13.3		5.0–21.6
20–39 (*n* = 61)		0.015	
BISIC group (*n* = 15)	19		12.1–25.9
CRS group (*n* = 46)	6.6		5.3–7.9

**Table 6 cancers-15-03401-t006:** Oncological effects of BISIC treatment.

BISIC Group	Pre-BISIC	Post-BISIC	Significance
PCI (mean ± SE)	27 ± 2.3	16 ± 2.1	0.003
PCI (median)	27	14	
Cytology			Conversion rate: 33%
positive	12	8	
negative	13	17	

## Data Availability

The data presented in this study are available on request from the corresponding author. The data are not publicly available due to institutional restrictions.

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
