# Peer review of "Benefit of Neoadjuvant Laparoscopic Hyperthermic Intraperitoneal Chemotherapy and Bidirectional Chemotherapy for Patients with Gastric Cancer with Peritoneal Carcinomatosis Considering Cytoreductive Surgery"

_cancers, 2023, doi:10.3390/cancers15133401_

Round 1

Reviewer 1 Report

The paper is extremely interesting, facing a challenging topic. The very good results obtained in terms of median and 5-years survival must be complimented, as well as early postoperative results, especially in the very advanced group of patients. 

MAJOR REVISIONS: As correctly admitted, limitations include the small patient number, the nonrandomized retrospective nature of the paper and lacking of propensity-score match analysis, that limits a lot statistical power. The p-value between the two groups in the multivariate analysis was only 0.049. 

Although overall morbidity rates are similar, it would be interesting to know the different complications , as well as mortality rates.

Operative time, blood loss and numer of tranfusions have been reported as negatively influencing the prognosis in the Literature, What do you attribute  your results ?

Tables 2 and 5 are quite confusing and wrong numbers have been included. In table 3 it not clear the meaning  and the relevance of the different median survival results. 

MINOR REVISION: two references are lacking in page 8. 

Statistical significance is low due to several bias, some methodologic errors and inaccurancies are evident. However, the topic is worth of investigation and results were promising for very advanced cases

Reviewer 2 Report

Yu HH et al. investigated the role of neoadjuvant laparoscopic HIPEC with bidirectional chemotherapy in stage 4 gastric cancer patients. They concluded that combined neoad HIPEC and bidirectional CHT is useful in PCI >10. 

Although the present study was conducted in a retrospective way, I believe it is properly conducted and deserves publication. I suggest a minor revision:

-       in abstract please simplify the sentence “than did direct CRS in terms of post-CRS peritoneal cancer index”

-       in introduction, language should be improved. Ex: “for PC originating from various cancer types [5]. The CRS is to eradicate macroscopic tumors, and the HIPEC is used to eliminate tiny or microscopic tumors.” or “with a large tumor burden remains poor, and CC in these patients is difficult.”

-       are you sure that “GCPC” is a necessary acronym?

-       “Even after curative surgery, PC recurrence affects nearly 46%...” do you mean AGC or stage 4 GC?

-       the biggest bias of this study is different pre-CRS bias. In limitation you wrote that, in spite of this,  BISIC group exhibited exhibited longer survival. Wouldn't it be appropriate to still perform a survival-only subanalysis comparing two groups with similar PCI?

-       Tumor burden evaluation should be explained in methods.

-       In discussion: French study demonstrated the superiority of CRS-HIPEC than CRS alone, while Yonemura introduced neoadjuvant IP and systemic chemotherapy. What about Italian study of Italian Peritoneal Surface Malignancies Oncoteam highlighted a significant difference in the survival rate according to neoadjuvant treatment (untreated patients: 10.7 months, 95% CI 5.1-16.2; treated patients: 35.3 months, 95% CI 2.8-67.8; p = 0.022). 10.1245/s10434-021-10157-0. I am not an author.

in introduction, language should be improved

Reviewer 3 Report

In this manuscript, the authors retrospectively reviewed 113 patients with gastric cancer with peritoneal carcinomatosis (GCPC), with 25 having received comprehensive treatment (L-HIPEC, BISIC, and then CRS-HIPEC; the BISIC group) and 88 having received direct CRS-HIPEC (the CRS group).

Various conclusions are drawn in the results:
- BISIC resulted in greater tumor clearance after CRS.
- BISIC significantly enhances overall survival (although MST after CRS did not differ significantly between the groups)
- Neoadjuvant BISIC is advisable for patients with an initial PCI of >10 due to improved MST.
- Oncological efficacy of BISIC based on conversion from malignant to non-malignant ascites and reduced PCI 

This is an extremely well written manuscript with sound description of methods, results and meaningful discussion.

Although the manuscript is based on an emerging management approach for GCPC and, as such, is very topical and relevant, it has limitations which may preclude it from publication in this very high impact factor journal. I do commend the authors for rightly describing those limitations of the manuscript/ research.

The main limitations of concern include the retrospective setting, low cohort numbers, the fact that one arm has only 25 patients and the single centre source of the cohort. These limitations greatly limit the conclusions that can be drawn from this research. Furthermore, the factors considered which resulted in some patients having received the comprehensive treatment (BISIC group) was not described, and as such the selection of patients to each treatment modality seems unexplained and a certain source of significant bias.

Certainly, this manuscript may be used as a prompt for further prospective, well matched, multi-centre trials in the future but in its current form I cannot endorse its publication in this particular journal.

Round 2

Reviewer 3 Report

As outlined in my previous report, there are several fundamental flaws with the study which limit the ability to draw meaningful conclusions.